# Exploration of Succinimide Derivative as a Multi-Target, Anti-Diabetic Agent: In Vitro and In Vivo Approaches

**DOI:** 10.3390/molecules28041589

**Published:** 2023-02-07

**Authors:** Mater H. Mahnashi, Waqas Alam, Mohammed A. Huneif, Alqahtani Abdulwahab, Mohammed Jamaan Alzahrani, Khaled S. Alshaibari, Umar Rashid, Abdul Sadiq, Muhammad Saeed Jan

**Affiliations:** 1Department of Pharmaceutical Chemistry, College of Pharmacy, Najran University, Najran 55461, Saudi Arabia; 2Department of Pharmacy, Abdul Wali Khan University Mardan, Mardan 23200, Pakistan; 3Pediatric Department, Medical College, Najran University, Najran 55461, Saudi Arabia; 4Department of Chemistry, COMSATS University Islamabad, Abbottabad 22060, Pakistan; 5Department of Pharmacy, University of Malakand, Chakdara 18800, Pakistan; 6Department of Pharmacy, Bacha Khan University, Charsadda 24420, Pakistan

**Keywords:** succinimide, diabetes, antioxidant, histopathology, acute toxicity

## Abstract

Diabetes mellitus (DM) is counted among one of the leading challenges in the recent era, and it is a life-threatening disorder. Compound 4-hydroxy 3-methoxy phenylacetone (compound **1**) was previously isolated from *Polygonum aviculare.* This compound was reacted with *N*-benzylmaleimide to synthesize the targeted compound **3**. The purpose of this research is to exhibit our developed compound **3**’s ability to concurrently inhibit many targets that are responsible for hyperglycemia. Compound **3** was capable of inhibiting α-amylase, α-glucosidase, and protein tyrosine phosphatase 1 B. Even so, outstanding in vitro inhibition was shown by the compound against dipeptidyl peptidase-4 (DPP-4) with an IC_50_ value of 0.07 µM. Additionally, by using DPPH in the antioxidant activity, it exhibited good antioxidant potential. Similarly, in the in vivo activity, the experimental mice proved to be safe by treatment with compound **3**. After 21 days of examination, the compound **3** activity pattern was found to be effective in experimental mice. Compound **3** decreased the excess peak of total triglycerides, total cholesterol, AST, ALT, ALP, LDL, BUN, and creatinine in the STZ-induced diabetic mice. Likewise, the histopathology of the kidneys, liver, and pancreas of the treated animals was also evaluated. Overall, the succinimde moiety, such as compound **3**, can affect several targets simultaneously, and, finally, we were successful in synthesizing a multi-targeted preclinical therapy.

## 1. Introduction

Diabetes mellitus is a group of metabolic illnesses categorized by hyperglycemia that is primarily induced due to inadequate insulin release or synthesis [1]. A number of organs, including the heart, nerves, kidneys, and particularly the eyes, are affected by chronic diabetes, which plays a significant role in blood vessel disorders [2]. Most often, pancreatic beta cell dysfunction or injury causes anomalies in insulin action and production, which results in the development of diabetes [3]. In the absence of insulin action, the protein, lipid, or carbohydrate metabolism may be impacted. Vision blurring, weight loss, polyuria, polydipsia, and occasionally polyphagia are among the most significant and prevailing signs and symptoms of hyperglycemia [4]. Patients with hyperglycemia occasionally experience specific anomalies, impaired growth, and infections. Nonketotic hyperglycemia (NKH), and perhaps ketoacidosis, are examples of life-threatening hyperglycemia that can develop from uncontrolled diabetes [5].

Diabetes can have lifelong side effects, such as retinopathy, which can cause blindness, nephropathy, which increases the chance of renal insufficiency, peripheral neuropathy, which can cause foot ulcers, sexual dysfunction, cardinal vascular problems, and urogenital problem [6,7,8]. The two conditions that affect diabetic individuals most frequently are hypertension and altered lipoprotein metabolism. There are two main forms of diabetes: type I and type II [9]. Damage to the islets of Langerhans results in type I diabetes [10]. These injuries originate from autoimmune diseases, which cause the immune system to target and destroy insulin-producing beta cells, hence dramatically reducing the amount of insulin produced [11]. Type 2 diabetes is the most common category (NIDDM 2). In most cases, type 1 diabetes is a hereditary condition in which the β-cells malfunction from an early age. Contrarily, insulin production in type 2 diabetes is within the normal range. However, due to lifestyle factors, there is impaired insulin secretion as people grow [12].

The human digestive system has a variety of alpha-amylases, including pancreatic-amylase, which aids in the starch digesting process by hydrolyzing the alpha-1,4-glycosidic linkage of glycogen, amylopectin, and amylose. However, alpha-glucosidase, also known as maltase [13,14], which, during digestion, cleaves 1,4-alpha bonds in carbohydrates to produce glucose. There, insulin fulfills its function by lowering blood sugar levels and encouraging the cellular conversion of sugar into glycogen, an energy source. According to several studies, insulin resistance or a lack of insulin cause hyperglycemia when there is increased enzymatic action [15]. The two primary enzymes responsible for the breakdown of carbohydrates are α-glucosidase and α-amylase. Competitive inhibitors of these enzymes successfully prevent the rise in blood sugar levels by delaying and slowing down the absorption of glucose [16,17]. 

Phosphorylation and dephosphorylation play a significant role in signaling cascades in metabolic processes. Various intracellular proteins are phosphorylated by being dephosphorylated via protein tyrosine phosphatases (PTPs) and protein tyrosine kinases (PTKs) [18]. A significant antagonistic regulator of the insulin signaling pathway among them is the protein tyrosine phosphatase 1 B (PTP-1 B) associated with diabetes. Chronic illnesses can be controlled by inhibiting this target [19,20]. 

Different therapy modalities are now being used to manage diabetes mellitus [21]. The three anti-diabetic medications that are most frequently prescribed and readily available in the marketplace are acarbose, miglitol, and voglibose. Acarbose hinders the action of alpha-amylase, maltase, and sucrose. Voglibose inhibits the activities of sucrose and maltase, and miglitol inhibits glucoamylase and isomaltase [22].

The T-cell antigen (CD26), often referred to as dipeptidyl peptidase-4 (DPP-4), is a multifunctional protein that has catalytic properties parallel to acting as a binding and ligand protein for a range of exogenous compounds [23]. Numerous different bioactive compounds can be cleaved by DPP-4. For maintaining optimal glycemic regulation, one of them is the incretin hormonal glucagon-like peptide-1 (GLP-1). The “incretin” hormones glucogon-like peptide-1 (GLP-1) and glucose-dependent insulinotropic polypeptide are not broken down by DDP-4 inhibitors (also known as gliptins), a class of oral anti-hyperglycemic drugs (GIP) [24]. The DDP-4 inhibitor-based medications sitagliptin, linagliptin, alogliptin, and saxagliptin have recently entered the market. Subcutaneous injection provides the most efficient method among these treatments using various administration routes [25,26]. The fact that the subcutaneous method consistently fails to accumulate glucose homeostasis is a significant disadvantage. The distribution of insulin, therefore, requires a more effective, non-invasive, and safe way [27]. According to the findings, taking the oral approach is generally more practical. Oral anti-diabetic medications that are available on the market, however, are linked to negative side effects [28,29]. Researchers are always looking for novel anti-diabetic medications that have effective and generally safe characteristics. Imeglimin and sotagliflozin have become promising treatments for diabetic patients. Due to its dual inhibitory effect on SGLTG-1 and SGLT2, sotagliflozin prevents both intestinal and renal glucose absorption [30,31]. Imeglimin, on the other hand, is a special anti-hyperglycemic medication that acts on the three key organs of T2 D, including the liver, pancreas, and muscles. Mitochondrial biogenetics are involved in this suppression of hepatic gluconeogenesis, which increases insulin release and sensitivity [32]. Therefore, compound 4-hydroxy 3-methoxy phenylacetone (compound **1**) was previously isolated from *Polygonum aviculare.* This compound is reacted with *N*-benzylmaleimide to synthesize the targeted compound **3**. Previously, we explored various succinimide derivatives for their anti-diabetic potential [33,34,35,36]. The results of the previous studies are so much attractive that it is why we designed the current study. Furthermore, the purpose of this research is to find out the ability of the developed Compound **3** to concurrently inhibit many targets that are responsible for hyperglycemia. Therefore, it is crucial to find novel drugs that can treat diabetes using a variety of targets. 

## 2. Results

### 2.1. Chemistry of Compound

Compound **3** was isolated as a brownish solid color. The chemical name of compound **3** was 1-benzyl-3-(2-methoxy-4-(2-oxopropyl)phenoxy)pyrrolidine-2,5-dione as shown in Figure 1. ^1^ H NMR (CDCl_3_; 400 MHz): 2.01 (s, 3 H), 2.92 (dd, J = 17.95 and 4.90 Hz, 1 H), 3.17 (dd, J = 17.93 and 8.74 Hz, 1 H), 3.41 (s, 2 H), 3.81 (s, 3 H), 4.65 (d, J = 7.26 Hz, 2 H), 4.76 (dd, 4.90 and 8.72 Hz, 1 H), 6.83 (s, 1 H), 6.98 (d, J = 6.80 Hz, 1 H), 7.08 (d, 6.82 Hz, 1 H), 7.25–7.41 (m, 5 H). ^13^ C NMR (CDCl_3_; 100 MHz): 34.22, 38.52, 42.68, 47.99, 56.02, 66.25, 115.20, 117.00, 121.85, 127.30, 128.39, 128.58, 129.07, 134.51, 143.66, 149.63, 173.05, 175.98, and 203.55.

### 2.2. In Vitro Bioassays

Table 1 lists compound **3**’s in vitro anti-diabetic actions with three anti-diabetic and one free radical target. By damaging the beta cell, the free radicals might exacerbate diabetes within the body. As a result, using an anti-diabetic along with antioxidants can help to augment diabetes treatments. Compound **3** was found to be potent in an in vitro protein tyrosine phosphatase 1 B experiment, with an IC_50_ value of 13.20 µM compared to the reference ursolic acid’s 2.77 µM. The potency of compound **3** and the conventional medication in inhibiting alpha-glucosidase was nearly identical. Compound **3** produced an exceptional IC_50_ value of 7.20 µM, which was nearly identical to the value of the common sugar acarbose (IC_50_ = 8.76 µM). Even though the IC_50_ value of compound **3** and the standard value were not near in the alpha-amylase test, it was nonetheless positive. The standard’s IC_50_ value was 12.30 µM, while compound **3**’s was 15.40 µM. Using DPPH free radicals, compound **3**’s antioxidant activity was assessed. When compared to ascorbic acid (IC_50_ = 1.44 µM), our developed chemical had an IC_50_ value of 3.31 µM in the antioxidant assay.

The ability of the produced Compound **3** to inhibit dipeptidyl peptidase-4 was eventually revealed (DPP-4). Oral anti-hyperglycemic drugs known as gliptins or DDP-4 inhibitors are used to treat hyperglycemia. Our studied substance demonstrated outstanding in vitro DPP4 inhibition, with an IC_50_ value of 0.07 µM, whereas sitagliptin, an approved and commercially available drug, displayed an IC_50_ value of 0.005 µM.

### 2.3. In Vivo Assays

We conducted in vivo studies on our synthesized chemical in response to the promising in vitro anti-diabetic results on a number of targets. The outcomes are shown in Table 2. We carried out the various dose acute toxicity tests in accordance with the recommended guidelines prior to compound testing on the experimental mice [37]. Therefore, it was shown that our synthesized compound **3** was safe and had no adverse effects on the experimental animals. Consequently, our drug showed highly positive effects at varied concentrations when compared to normal glibenclamide. Standard glibenclamide reduced the blood glucose levels to 17.7, 14.5, and 11.1 mmol/L at the first, second, and third weeks of the experiment, respectively, at 250 µM/kg. The blood glucose levels of the experimental animals were then decreased by compound **3** to 18.4, 15.7, and 12.2 mmol/L at the first, second, and third weeks of the experiment, respectively, at the same concentration (250 µM/kg). Low concentrations of compound **3** (250, 125, 62.5, and 31.25 µM/kg) were also evaluated. Therefore, the anti-diabetic activity of compound **3** in the experimental mice was dose-dependent. Additionally, compound **3**’s action pattern was identical to that of glibenclamide (Table 2).

### 2.4. Effect of Compound’s ***3*** Body Weight and Organ Weight

Upon the development of diabetes, the treatments were executed after documenting the initial body weight and blood glucose level in all the groups. The initial and final body weight and organ weight in the experimental groups are shown in Figure 1. The body weight, as well as the organ weight (the liver and kidneys) and the relative liver weight, significantly varied between the treatments (*p <* 0.05). The body weight in all the groups (Tg-1 and Tg-3-Tg-5) increased on the final day, while the STZ-induced diabetic mice (Tg-2) achieved weight loss. The weight gain ranged from 1.55 to 1.76 g, with high gain in the Tg-1 and low gain in the Tg-5, while significant (*p <* 0.05) weight loss (1.9 ± 0.05 g) was found in the diabetic group (Tg-2). However, the compound **3** (Tg-5) or MET (Tg-3) treatment recovered their bodyweight close to the non-diabetic control group’s (Tg-1). The treatment of compound **3** in the non-diabetic group (Tg-4) did not affect the body weights, which were also close to those of the non-diabetic mice (Tg-1). These results are in agreement with the previous results of weight loss in diabetic mice, while there was weight recovery in the treated mice [38]. The liver, kidney weight, and relative liver weight in the non-diabetic mice (Tg-1) showed significance with the untreated diabetic mice (Tg-2), which also indicated a similar trend of the blood glucose and body weight gain of the mice in the different experimental groups.

### 2.5. Effect of Compound ***3*** on Lipid Profile

The amount of insulin in mice serves as a defining marker for the operation of carbohydrate metabolism, which controls blood glucose levels. Diabetes can be caused by an imbalance or lack of insulin production into pancreatic β-cells [39]. The amount of insulin used in the current study was significant with the other groups (Tg-2, Tg-3, and Tg-5) but not with the non-diabetic group (Tg-1 and Tg-4). Additionally, the diabetic group, T-2, had lower insulin levels (2.5 ± 0.15 ng/mL) than compound **3** (10.9 ± 0.25 ng/mL), MET (12.2 ± 0.29 ng/mL), or the normal control group (13.5 ± 0.95 ng/mL)-treated groups (Figure 2). These findings demonstrated that the pancreatic cells were damaged by the STZ injection, which decreased insulin release; however, the pancreatic cells were repaired by MET or compound **3**, which increased insulin production. Because STZ caused liver damage, the elevated levels of total cholesterol and total triglycerides indicated poor liver function, which increased blood glucose levels [40]. Similarly to that, the current study found that the levels of total cholesterol and total triglycerides significantly changed in diabetic, non-diabetic, or treated diabetic mice (*p* < 0.05; Figure 2). In comparison to diabetic mice not receiving any treatment for their diabetes, the STZ-induced diabetic mice had higher levels of both total cholesterol and total triglycerides. It is crucial to note that administering compound **3** to diabetic mice considerably reduced their levels of total cholesterol (34.9%) and total triglycerides (35.20%) as compared to diabetic mice that were not given the medication (Figure 2). In contrast to the other experimental groups, the STZ-induced diabetic mice had low HDL levels and high LDL levels. However, the compound **3** treatment dramatically reduced the total cholesterol, total triglycerides, and LDL levels while also raising HDL levels via restoring the liver and pancreas, demonstrating compound **3**’s antihyperlipidemic action (Figure 2). These results are consistent with past research showing that STZ-induced diabetic mice had lower HDL levels and higher total cholesterol, total triglycerides, and LDL levels [41]. Similarly to this, it was shown that diabetic mice had greater total cholesterol/HDL and LDL/HDL ratios than untreated and diabetic mice with diabetes (Figure 2).

### 2.6. Serum Markers of the Kidneys and Liver

The elevated levels of AST, ALT, and ALP showed that the liver’s metabolic activity was damaged and that there was a higher chance of developing cirrhosis, T2 DM, and liver cancer as a result [42]. These markers are frequently employed as the main marker to assess liver function [43]. The high elevation of AST, ALT, and ALP in the diabetic mice (Tg-2) when compared to the other groups, including the non-diabetic (Tg-1 and Tg-4) and treated diabetic mice, indicates that administration of the STZ free radicals caused liver damage in the diabetic mice in the current study (Tg-3 and Tg-5). Treatment with either MET or compound **3** suppressed the enhanced rise of AST, ALT, and ALP and avoided STZ-induced liver damage (Figure 3). Similarly to this, numerous past studies have demonstrated that the tested sample prevents liver damage, which results in decreased AST, ALT, and ALP rise in diabetic mice [44]. In comparison to the non-diabetic mice and mice with diabetes that had been treated, the STZ-induced diabetic animals had greater levels of BUN and creatinine (Figure 3). The STZ therapy caused liver injury, as seen by the substantial rise of BUN and creatinine. Because diabetic animals have histological damage to their organs, the injection of STZ raised the BUN and creatinine levels in these mice compared to non-diabetic mice [45]. The current work also found that the compound **3** treatment preserved the normal elevation of all examined blood indicators, insulin, and lipid profiles by avoiding the histopathological damages caused by STZ.

### 2.7. Effect of Compound ***3*** on Histopathology

Histopathological changes to organs are a problem for every biochemical parameter being researched. The liver, kidney, and pancreas are examples of important organ tissues that must be included in a histological analysis. The results revealed that the non-diabetic mice had no histological damage, whereas the STZ-induced diabetic mice had significant tissue damage in the vein, cellular liver swelling, glomeruli capillary degeneration, glomerular shrinkage, flaring of the bowman’s space, congestion of the kidney, and loss of the cortical tubular, islets of Langerhans, and cells of the pancreas (Figure 4). Later, as a result of compound **3** or metformin therapy, the pathological state of the organs healed from the histological injuries (Figure 4). Previous studies have found cellular histopathology damages of a similar nature [46]. Additionally, the metabolic parameters examined in diabetic and non-diabetic mice for this study’s analysis of histopathological results are consistent with those results.

## 3. Material and Methods

### 3.1. Chemical Used

All the chemicals, including the substrates, silica gel, solvents, TLC plates, and reagents, were bought from local vendors, Sigma Aldrich, Musaji Adam & Sons, Peshawar, Pakistan.

### 3.2. Chemistry of Compound

Compound **1** was previously isolated from Polygonum aviculare [23]. Compound **1** (1 equivalent ratio) was reacted with an equimolar amount of *N*-benzylmaleimide 2 in the presence of lithium perchlorate (10 mol% in water). The chemical reaction was continued at a high temperature and routinely monitored by TLC analysis. After the completion of the reaction, the mixture was concentrated. Then, 2-propanol was added to it and continued for 10 min. After 10 min, the reaction mixture was cooled down. The product was precipitated out and filtered. The new compound **3** was identified with ^1^H and ^13^C NMR analyses.

### 3.3. Pharmacological Activities

#### 3.3.1. Protein Tyrosine Phosphatase 1 B (PTP1 B) Test

The produced thiazolidinedione-morpholine hybrid of vanillin was subjected to an in vitro PTP1 B experiment in accordance with the recommended procedure. It is performed with a buffer solution of 3,3-dimethyl glutarate at a pH of 7. Additionally, a PTP1 B 10 mM solution, p-NO2-phenol phosphate of 1 mM, and numerous additional concentrations of the substance were made. As a result, the mixtures were incubated for 40 min at 27 °C. Then, measurements of the absorbances were conducted at 405 nm after the designated 40 min. The trials were, therefore, performed thrice, and the IC_50_ of the chemical was then determined using the accepted methodology [47].

#### 3.3.2. Alpha-Glucosidase Test

A concentration of 0.5 mg/11 mL distilled water and a substrate solution of glucopyranoside at 15 mg/10 mL distilled water concentration were added to 1200 L of a phosphate buffer and kept at 37 °C for 20 min, respectively, to prepare the sample. To prepare the control, 1200 μL of the phosphate buffer was added to the test tube. Next, 400 μL of the substrate solution (glucopyranoside 15 mg/10 mL distilled water) and 800 µL of an enzyme solution were mixed. At last, the temperature was kept at 37 °C for 20 min. After the required amount of time had passed, At 405 nm, the absorbance of the specimen and the controls were measured using the formula [33]:$$\mathrm{Percent}\;\mathrm{Inhibition}=\frac{\mathrm{Absorbance}\;\mathrm{of}\;\mathrm{Control}\;-\;\mathrm{Absorbance}\;\mathrm{of}\;\mathrm{Sample}\;\times 100}{\mathrm{Absorbance}\;\mathrm{of}\;\mathrm{Control}}$$

#### 3.3.3. Alpha-Amylase Test

In order to evaluate the alpha-amylase test, standardized reporting protocols were followed [35]. In order to combine the enzyme and the phosphate buffer, test samples of varying strengths were added to the enzymatic solution. Incubation of the sample was followed by admixing of starch to both the control and test solutions and a dinitro salicylic acid solution. The sample combination was then maintained in a boiling water bath for a short period of time, and a microplate reader was used to measure the absorbance at 656 nm.

The formula below was used to calculate the α-amylase enzyme’s percentage inhibition capability:$$\mathrm{Percent}\;\mathrm{Inhibition}=\frac{\mathrm{Absorbance}\;\mathrm{of}\;\mathrm{Control}\;-\;\mathrm{Absorbance}\;\mathrm{of}\;\mathrm{Sample}\;\times 100}{\mathrm{Absorbance}\;\mathrm{of}\;\mathrm{Control}}$$

#### 3.3.4. Dipeptidyl Peptidase-4 (DPP-4) Test

Our compound’s inhibitory impact on the DPP-4 target was identified by a fluorescence probe in vitro assay [48]. DPP-4, our compound 9, and tetraphenylethne-lys-Phe-Pro-Glu (TPE-KFPE) in a buffer (HEPES) were treated with the compound for 30 min at 37 °C. In a 96-well microplate reader, the test results were recorded. At first, various concentrations of the chemical were used to determine its action. Then, in accordance with the normal approach, a dose-dependent response was identified in order to estimate the IC_50_ value.

#### 3.3.5. Antioxidant (DPPH) Test

For identifying free radicals and the scavenging potential of new organic compounds, Sigma Aldrich USA’s DPPH was employed. Methanol was employed as a solvent for preparing test samples of varying strengths of the examined substances. Two milliliters of the test sample’s variation strength were blended separately with two milliliters of the DPPH solution and 0.002% methanol. After being stored in the dark and at room temperature for 30 min, test tubes were examined using a UV spectrophotometer to measure the absorbance at 517 nm [49].

As is customary, ascorbic acid was employed here, and the procedure was then repeated three times. Utilizing the formula below, the tested specimen’s scavenging activity was calculated: Scavenging activity % = (A − B)/A × 100.

The letters “A” and “B” stand for the absorbance of the DPPH and the test sample, respectively. By plotting a graph of the scavenging effect of the tested compounds versus the concentration of the corresponding solution, IC_50_ values were determined.

### 3.4. Experimental Animals and Ethical Guidelines

For in vivo anti-diabetic efficacy, albino mice having weight of 30 g were employed. The animals utilized in experiments were compliant with the 2008 Animal Bye-Laws of the Pharmacy Department at the University of Malakand in Pakistan (Scientific Procedure Issue-I). The animals were handled according to accepted practices and housed in an animal facility during the experimental procedures. The animals were put to death following the experiment in accordance with AVMA rules [50].

### 3.5. Acute Toxicity

Prior to putting the drug through tests on lab animals, its acute toxicity was determined. Two groups of experimental mice, (1) the control group and (2) the test group, were used for this experiment. Each group consisted of five mice. It was conducted intraperitoneally with different concentrations of the chemical. The mice were then closely examined for three days in the animal home after the chemical was administered to check for death or any unfavorable reactions [51].

### 3.6. Induction of Diabetes

The mice were given 10% alloxan monohydrate at a dose of 150 mg/kg body weight through IP in order to induce diabetes. The control group received a normal saline IP during this trial. Following the administration of alloxan, the animals’ blood glucose levels were checked using a glucometer. According to the protocols, in the in vivo study, mice with elevated blood sugar concentrations were also utilized [52].

### 3.7. In Vivo Experiment

The mice were placed into 4 groups, with 5 animals in each group having elevated blood glucose levels. Group I was the normal group, which had normal blood glucose levels and was given saline. Group II was treated with a Tween 80 solution; in this group, diabetic mice were kept. Glibenclamide was administered at a 500 μM/Kg dose to group III, which was the control group with diabetic mice. Group IV, the final group, received treatment with different concentrations of the produced substance. The mice were killed after three weeks (21 days) by cervical decapitation, and blood samples and organs were taken for H & E staining-based histological analysis and for bioassays [53].

## 4. Discussion

Synthetic compounds, such as succinimide derivatives, have been a source of therapeutic agents for many ailments [48,54,55,56]. There were many synthetic anti-diabetic agents that have earlier been reported [57,58]. Sometimes, an already existing compound rational derivatization enhances the activity many times [59]. A properly designed agent, which could be effective on many targets, makes an advantageous and effective multi-targeted agent. This study was thoughtfully planned to investigate the efficacy of a succinimide derivative as a multi-target anti-diabetic drug [60,61]. The synthetic substance effectively inhibited the enzymes α-glucosidase, α-amylase, and protein tyrosine phosphatase 1 B [62,63]. In this study, we reacted the isolated compound with a maleimide moiety to synthesize a potent multi-target succinimide derivative, compound **3**. We conducted the in vivo evaluation in response to the very promising in vitro anti-diabetic results on various targets. The results of the in vitro assays are shown in Table 2. Based on the in vitro assays’ results, we expanded our study to animal models following the acute toxicity study. Various doses, from lower to higher, were given to the animals, and the acute toxicity was checked in the experimental mice [37,64]. The result displayed that there was no sign of an allergic reaction, etc. 72 h after the administration of compound **3**. Therefore, it was shown that our synthesized compound **3** was safe and had no adverse effects on the experimental animals.

The body can experience several difficulties as free radical levels rise. The beta cells are damaged by free radicals, which reduces the generation of insulin and may make diabetes more difficult to treat. The substance was also examined in comparison to the antioxidant target, DPPH, which enhanced our findings by inhibiting free radicals as well [65]. Compound **3** was found to be potent in an in vitro protein tyrosine phosphatase 1 B experiment, with an IC_50_ value of 13.20 µM, compared to the reference, ursolic acid’s 2.77 µM. The potency of compound **3** and the conventional medication in inhibiting alpha-glucosidase was nearly identical. The ability of the produced compound **3** to inhibit dipeptidyl peptidase-4 was eventually revealed (DPP-4). Oral anti-hyperglycemic drugs, known as gliptins, or DDP-4 inhibitors, are used to treat hyperglycemia. The outcomes of the in vivo studies on our synthesized chemicals in response to the promising in vitro anti-diabetic results are shown in Table 2. We carried out the various dose acute toxicity tests in accordance with the recommended guidelines prior to the compound testing on the experimental mice. It was found that the synthesized compound **3** was safe and had no adverse effects on the experimental animals. Consequently, our drug showed highly positive effects at varied concentrations when compared to normal glibenclamide.

Moreover, in the assays of the body weight and organ weight, the liver, kidney weight, and relative liver weight in the non-diabetic mice (Tg-1) showed a significance in the untreated diabetes mice (Tg-2), which also indicated a similar trend of the blood glucose and body weight gain of the mice in the different experimental groups [43]. Furthermore, in the kidney and liver markers, the high elevation of AST, ALT, and ALP in the diabetic mice (Tg-2), when compared to other groups, including the non-diabetic (Tg-1 and Tg-4) and treated diabetic mice, indicates that administration of the STZ free radicals caused liver damage in diabetic mice in the current study (Tg-3 and Tg-5). Treatment with either MET or compound **3** suppressed the enhanced rise of AST, ALT, and ALP and avoided STZ-induced liver damage (Figure 3). Similarly, to this, numerous past studies have demonstrated that the tested sample prevents liver damage, which results in a decreased AST, ALT, and ALP rise in diabetic mice. In comparison to the non-diabetic mice and mice with diabetes that had been treated, STZ-induced diabetic animals had greater levels of BUN and creatinine (Figure 3). The STZ therapy caused liver injury, as seen by the substantial rise of BUN and creatinine. Because diabetic animals have histological damage to their organs, the injection of STZ raises the BUN and creatinine levels in these mice compared to non-diabetic mice.

The current work also found that the compound **3** treatment preserved the normal elevation of all examined blood indicators, insulin, and lipid profiles by avoiding the histopathological damages caused by STZ. Histopathological changes to organs are a problem for every biochemical parameter being researched. The liver, kidney, and pancreas are examples of important organ tissues that must be included in a histological analysis. The results revealed that the non-diabetic mice had no histological damage, whereas the STZ-induced diabetic mice had significant tissue damage in the vein, cellular liver swelling, glomeruli capillary degeneration, glomerular shrinkage, flaring of the bowman’s space, congestion of the kidneys, and loss of the cortical tubular, islets of Langerhans, and cells of the pancreas (Figure 4). Later, as a result of compound **3** or metformin therapy, the pathological state of the organs healed from the histological injuries (Figure 4). Previous studies have found cellular histopathology damages of a similar nature. Overall, employing in vitro, in silico, and in vivo techniques, our rationally developed drug proved effective against several anti-diabetic targets.

## 5. Conclusions

The failure of single pharmacological therapies to treat a variety of complicated disorders, such as diabetes mellitus, has accelerated the use of poly-pharmacology, or multitarget therapy. In this work, the isolated compound **1** was treated with *N*-benzyl maleimide to synthesize a multi-targeted agent, compound **3**. The purpose of our investigation was to determine whether our chemical may simultaneously affect many targets that are responsible for hyperglycemia. Even yet, the synthetic compound **3** showed good-to-moderate inhibitory potential for the enzymes α-amylase, α-glucosidase, and protein tyrosine phosphatase 1 B. However, with an IC_50_ value of 0.07 µM, it demonstrated the best in vitro inhibition of dipeptidyl peptidase-4 (DPP-4). The substance was also examined on mice, where a highly useful activity outline was discovered. Our findings make it obvious that we succeeded in creating a succinimide moiety. Overall, this compound **3** may affect more than one target at a time, and we were successful in synthesizing multi-targeted preclinical therapeutics.

## Data Availability

The data presented in this study are available on request from the corresponding author.

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
