# Peer review of "Exploration of Succinimide Derivative as a Multi-Target, Anti-Diabetic Agent: In Vitro and In Vivo Approaches"

_molecules, 2023, doi:10.3390/molecules28041589_

Round 1
Reviewer 1 Report
The manuscript by Mahnashi et al. titled "Exploration of Succinimide Derivative as Multi-Target Anti-Diabetic Agent: Invitro and Invivo Approaches" describes an anti-Diabetic agent of Succinimide derivatives. The approach is discussed in the invitro and invivo studies. The experimental method is very appropriate, and the results discuss in detail. I believe that it will help scientists who work with diabetics in the future. I recommend the manuscript for publication after minor English corrections.
Author Response
Reviewer 1:
- I recommend the manuscript for publication after minor English corrections.
Reply: Thank you so much for your comments and vigilant review. The manuscript was revised thoroughly for English corrections; all the mistakes were removed and highlighted.

Reviewer 2 Report
1. This manuscript has done a lot of research work, and the research is innovative. Therefore, I think this manuscript can be accepted after appropriate modification. But there are some small problems.
2. The introduction of the manuscript does not explain why this research is needed, and it is not logical.
3. The discussion part of the manuscript is too simple, and there is no in-depth discussion around the research results.
Author Response
Reviewer 2:
- This manuscript has done a lot of research work, and the research is innovative. Therefore, I think this manuscript can be accepted after appropriate modification. But there are some small problems.
Reply: Thanks to the worthy reviewer comments and suggestion. The manuscript is checked thoroughly, and all the problems were properly corrected one by one and highlighted.
- The introduction of the manuscript does not explain why this research is needed, and it is not logical.
Reply: Thanks for the reviewer comments, the needful suggested changes have been done and highlighted.
- The discussion part of the manuscript is too simple, and there is no in-depth discussion around the research results.
Reply: Thank you so much for such a vigilant review, the discussion section is updated in the revised manuscript.
